# Stability and sensitivity of interacting fermionic superfluids to quenched disorder

Jennifer Koch [1,2], Sian Barbosa [1], Felix Lang [1] & Artur Widera [1]✉

The microscopic pair structure of superfluids has profound consequences on their properties. Delocalized pairs are predicted to be less affected by static disorder than localized pairs. Ultracold gases allow tuning the pair size via interactions, where for resonant interaction superfluids show largest critical velocity, i.e., stability against perturbations. The sensitivity of such fluids to strong, time-dependent disorder is less explored. Here, we investigate ultracold, interacting Fermi gases across various interaction regimes after rapid switching optical disorder potentials. We record the ability for quantum hydrodynamic expansion of the gas to quantify its long-range phase coherence. Contrary to static expectations, the Bose-Einstein condensate (BEC) exhibits significant resilience against disorder quenches, while the resonantly interacting Fermi gas permanently loses quantum hydrodynamics. Our findings suggest an additional absorption channel perturbing the resonantly interacting gas as pairs can be directly affected by the disorder quench.

When an interacting system of fermionic particles is cooled below a critical temperature, bosonic pairs form, and the system becomes superfluid or, for charged fermions, superconducting. For ultracold Fermi gases, magnetic Feshbach resonances allow the modification of the effective interaction strength[1] and thereby, the underlying pairs from small, bound molecules to delocalized Cooper-type fermion pairs along the so-called crossover from a molecular Bose-Einstein condensate (BEC) to a Bardeen-Cooper-Schrieffer (BCS) superfluid[2,3]. Between these two regimes, on resonance, a unitary Fermi gas (UFG) is realized, exhibiting, for example, the largest superfluid critical velocity along the crossover[4,5]. On resonance, the microscopic details of the gas are not relevant for the macroscopic properties[6], which has allowed us to deduce universal properties highly relevant for nuclear matter, for example[2,3,7–10]. The microscopic pairing has been investigated theoretically[11,12] and experimentally using radio-frequency (rf) spectroscopy, for example, revealing different equilibrium properties such as the pair size or binding energy[13]. This microscopic pairing has profound consequences on the macroscopic quantum state. The excitation spectrum, for example, has been measured along the crossover in three-dimensional gases using Bragg spectroscopy[14,15] and rf spectroscopy[16] and shows a clear change from a phononic-type branch in the BEC regime to a spectrum indicating dominant pair breaking in the BCS regime.

A prominent consequence of the paired nature in disordered potentials is summarized by the Anderson theorem[17], stating that delocalized Cooper pairs are only little affected by local perturbations, leading to only a moderate decrease of pairs for disordered systems due to a disorder-induced reduction of the density of states close to the Fermi edge. Indeed, theoretical investigations of the BEC-BCS crossover in static disorder[18–22] show only a slight modification of the critical temperature in the BCS regime[18], together with a relatively small reduction of the order parameter and condensate fraction[19,20], and an area of stability in the crossover region close to resonant interactions. Importantly, most studies have considered the weak disorder regime, and theoretical investigations of strong-disorder systems beyond perturbation theory are just emerging[23]. Experimentally, for strong interactions, the emergence of a fragmented Fermi gas has been observed in static disorder[24]. In our experiment, we consider the effect of strong and time-dependent disorder.

By contrast, in the BEC regime, the critical temperature is more strongly suppressed by static disorder, together with order parameters and condensate fraction. Experimental transport measurements indicated a disorder-induced transition from a superfluid to a

[1]Department of Physics and Research Center OPTIMAS, RPTU Kaiserslautern-Landau, Kaiserslautern, Germany. [2]TOPTICA Photonics AG, Gräfelfing, Germany. ✉e-mail: widera@rptu.de

normal fluid in strong disorder[25]. The different behavior in the two regimes can be well explained by the fact that strong local interactions increase the ability of the superfluid to react to local perturbations, drawing the picture of the UFG being the most resilient superfluid. However, the specific properties of the pairs are expected to also modify the response of the superfluid to *time-dependent* disorder and determine how fast quantum properties decay for strong perturbation, or how fast they are recovered once the perturbation is absent. Experimental studies along the BEC-BCS crossover in the regime of static disorder regime are scarce, however, it has been shown, for example, the damping coefficient of dipole oscillations in a static disorder potential depends on the interaction strength[26].

In this work, we probe the response of ultracold, interacting Fermi gases of lithium atoms in the BEC-BCS crossover to strong perturbations in space and time via rapidly switched optical disorder potentials with a focus on the BEC side. We measure the time evolution of long-range coherence quantified via the ability of the gas to expand hydrodynamically. We find that, in marked contrast to the expectation of the static disorder or weak-perturbation case, the quantum properties of a UFG are more strongly suppressed than a molecular BEC (mBEC), indicating an increased sensitivity. For quenches out of a disordered potential, we find that the UFG never restores quantum hydrodynamics for all parameters investigated, while an mBEC re-establishes quantum hydrodynamics, even when the quench leads to strong particle losses of up to 70%. Temperature measurements indicate an additional heating channel specific for gases close to resonant interactions, leading to strong local dephasing or pair breaking.

## Results

### Experimental realization

We experimentally study the response of an ultracold gas of fermionic lithium-6 (Li) atoms prepared in the two lowest Zeeman sub-states (typically $N_i = 3 \cdot 10^5$ atoms per spin state $i = \uparrow, \downarrow$) to fast switching of a disorder potential. The gas is prepared at a magnetic field of 763.6 G and has a temperature $T \approx 200$ nK, corresponding to $T/T_F = 0.3$ in the BEC regime at 680 G. Typical values for trap frequencies are $\omega_x, \omega_y, \omega_z = 2\pi \times (345, 23, 220)$ Hz (see Methods). To tune the interaction between the two internal states, and hence the pair size of the superfluids formed, a broad Feshbach resonance at a magnetic field of 832.2 G is used[27]. In order to prepare a different interaction regime, we adiabatically ramp the magnetic field to the desired final value. Due to the ramp, the reduced temperature drops to values well below $T/T_F < 0.17$ at unitarity[28], implying $T < T_c$ with the critical temperature $T_c$[29]. This allows entering the different regimes of the BEC-BCS crossover, quantified via the interaction parameter $1/k_F a$, with the absolute value of the Fermi wave vector

$$k_F = \sqrt{2mE_F}/\hbar \qquad (1)$$

with the mass $m$ of a Li atom and the Fermi energy[3]

$$E_F = \hbar\bar{\omega}(3N_{\uparrow\downarrow})^{1/3}, \qquad (2)$$

the s-wave scattering length $a$, the geometric mean $\bar{\omega}$ of the trap frequencies, and the total atom number $N_{\uparrow\downarrow}$ in both spin states. The typical value for the Fermi energy is $E_F \approx k_B \times 670$ nK, with $k_B$ the Boltzmann constant. Table 1 shows typical experimental parameters for the quantum gas at different magnetic fields.

Our observable revealing the response of quantum properties is the ability of the gas to undergo quantum hydrodynamic expansion. Expansion measurements probing hydrodynamic behavior have been used, for example, to characterize the transition between ballistic and hydrodynamic expansion[30] and to study the viscosity in a UFG[31]. Here, we employ hydrodynamic expansion as a measure of long-range phase coherence[32]. It allows us to time-resolve a quantum system's response

**Table 1 | Typical characteristic parameters of the quantum gas for three specific magnetic field values**

| magnetic field $B$ (G) | s-wave scattering length $a$ ($a_0$) | Fermi wave vector $k_F$ ($\mu m^{-1}$) | peak density $n_0$ ($10^{12}$cm$^{-3}$) |
|---|---|---|---|
| 680.0 | 1238 | 4.02 | 11.3 |
| 763.6 | 4509 | 4.06 | 5.3 |
| 832.2 | 1758077 | 4.09 | 3.8 |

We extract the values of the s-wave scattering length $a$ from[27], and they are given in units of the Bohr radius $a_0$. The wave vector is computed according to Eq. (1) with the Fermi energy Eq. (2). The densities $n_0$ given are peak densities in the center of the trap.

to a strong perturbation in space and time, revealing the existence or reformation of a well-defined global phase. We can thus trace a genuine quantum property of a strongly interacting, three-dimensional many-body system subjected to strong and time-dependent perturbation. In a previous work[32], performed with the same experimental apparatus, we focused on investigations deep in the mBEC regime. Here, we extend this study by focussing on the markedly different stability and sensitivity of an mBEC compared to a UFG.

Hydrodynamics is initiated by suddenly switching off the optical dipole trap (see Fig. 1a), releasing the gas into a magnetic trap with a saddle potential configuration (confining in the x-y plane and anti-confining in the z-direction), where the change of aspect ratio $A$ can be determined from absorption images. Here, the aspect ratio is defined as the ratio of the full width at half maximum (FWHM) of the 1D integrated column density distributions in $x$ and $y$ directions (see Fig. 1a and Supplementary Note 1A). The ensuing dynamics are markedly different for an ideal gas, a classical (thermal) interacting gas or a quantum gas[33]. While an ideal gas shows a moderate change in aspect ratio due to single-particle dynamics in the trap, this change is enhanced by collisional hydrodynamics for repulsively interacting classical gases (see Fig. 1b, d, e). Quantum gases, however, show a remarkably large change in aspect ratio, as shown in Fig. 1b, c, e. This strong increase beyond classical collisional hydrodynamics is tightly connected to the existence of a well-defined global phase, i.e., long-range phase coherence[32]. We note that a comparable quantum enhancement cannot be observed for the BCS side of the resonance, i.e., negative interaction parameters $1/k_F a < 0$ even close to resonance, in agreement with previous observations of the vanishing of hydrodynamics when approaching the BCS regime[34]. We attribute this to the breaking of the underlying fermionic pairs due to the strongly reduced density and, hence, paring gap during the expansion. Our analysis is therefore restricted to positive interaction parameters $1/k_F a \geq 0$ which will be the focus throughout the remainder of this work. For more details of the experimental setup, see Methods and ref. 35.

The perturbation of the system is controlled through a repulsive optical speckle disorder-potential with mean potential strength $\bar{V}_{dis}$ and correlation lengths of $\sigma_{x,y} = 750$ nm, and $\sigma_z = 10$ $\mu$m, see Fig. 1a. Here, the correlation length is defined as the $1/e$ − width of the speckle pattern's autocorrelation function and quantifies the typical grain size[32]. The mean potential strength is of the order of the superfluid's chemical potential $\mu$ ($\mu_{BEC} = k_B \times 390$ nK for a mBEC and $\mu_{UFG} = k_B \times 480$ nK for a UFG) and the short correlation length is larger but of the order of the quantum gases' coherence or healing length $\xi$ ($\xi_{heal} \approx 230$ nm for $1/k_F a \approx 1$). Thus, the static perturbation is strong. Since the polarizability of a Feshbach molecule is twice as high as that of a single atom, the potential height of the speckle potential for the molecules is twice as high as for the unbound atoms at the same laser power[36,37]. In tunneling experiments along the BEC-BCS crossover, for example, this does not seem to play a role, and the dynamics for constant optical potentials along the crossover could be compared[38,39]. Measurements for atom losses after disorder quenches (see Supplementary Note 1B) show, however, that the loss curves collapse to a single curve, when the disorder laser power for the UFG is

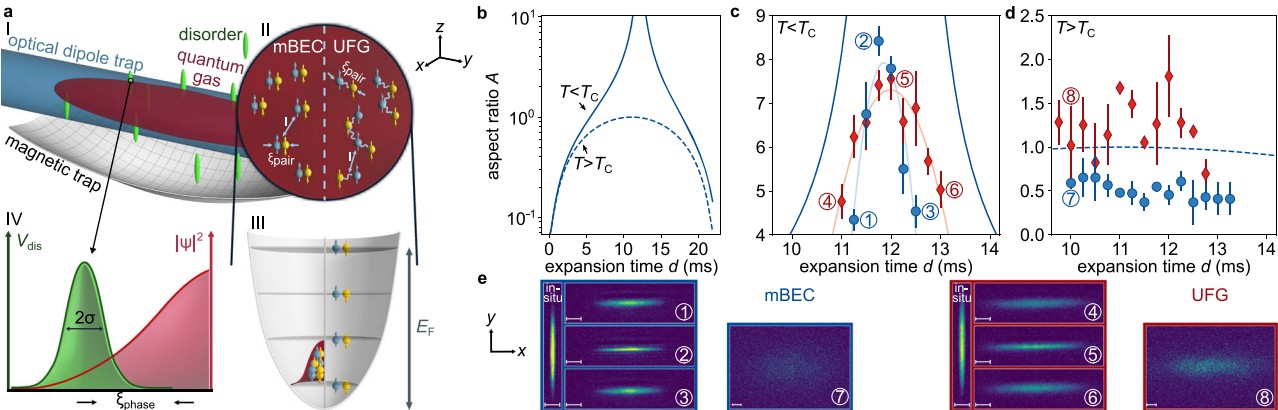

**Fig. 1 | Experimental setup and method. a** (I) Schematic drawing of the experimental setup. The atomic cloud (red ellipsoid) is trapped in a combination of an optical dipole trap (blue cylinder) and a magnetic saddle potential (gray surface) and is superposed by the disorder field (green ellipsoids). (II) The microscopic pair size $\xi_{\text{pair}}$ of an mBEC is much smaller than that of a UFG, and the pair distance $l$ equals roughly the pair size at resonance. (III) In the mBEC regime, the Feshbach molecules occupy the ground state of the harmonic trap and form a macroscopic wavefunction; by contrast, in a UFG, the atoms occupy the levels according to the Pauli exclusion principle up to the Fermi energy $E_{\text{F}}$. (IV) The microscopically relevant length scale is the coherence (healing) length $\xi_{\text{phase}}$ quantifying the length on which the system can respond as a macroscopic many-body wavefunction to perturbations such as disorder speckles. **b** Computed ideal time evolution of the aspect ratio for a degenerate gas (solid line) and a thermal gas (dashed line) for hydrodynamic expansion in our experimental traps without disorder, showing a divergence of the aspect ratio for a quantum fluid. For details, see Methods. **c** Measured aspect ratio $A$ of an mBEC at 763.6 G (blue circles) and a UFG at 832.2 G (red diamonds) as a function of the expansion time $d$ around the maximum achievable value, which is limited by experimental details, see ref. 32. Each point is an average of four repetitions, and the corresponding error bars show their standard deviations. The lines (light blue and light red) are guides to the eye. The solid dark blue line is equivalent to the computed ideal time evolution of the aspect ratio in (**b**) for a degenerate gas. **d** The same measurement as in (**c**) but for thermal gases. The dashed blue line is equivalent to the computed ideal time evolution of the aspect ratio in (**b**) for a thermal gas. **e** Absorption pictures for the measurement points indicated in panels (**c**) and (**d**), where the left most pictures show the initial quantum gases before expansion dynamics. The scales in the images mark a distance of 50 μm, also in the following figures.

twice the power for a mBEC, supporting our assumption of different potentials $V^{(\text{mBEC})} \approx 2\,V^{(\text{UFG})}$ for the same laser power. This observation indicates that the disorder locally affects molecules in the mBEC and atoms in the UFG. In the following, the disorder is given in laser power of the disorder, providing a comparison of potential heights differing by a factor of two. Moreover, the disorder potential can be either ramped adiabatically with respect to $h/\mu$ or be quickly switched on or off with switching times being shorter than $h/\mu$ (see insets of Figs. 2a, 3a), so that the many-body system cannot adiabatically follow the dynamics. The speckle realization is changed after each measurement. Furthermore, for our parameters, classical trapping cannot occur, because the mobility edge is close to the classical percolation threshold in our system[36,40]. For a detailed description and characterization of the disorder potential see Methods and Refs. 23,41,42.

## Decay of quantum hydrodynamics

We first study the impact of a suddenly applied disorder potential on a superfluid in different interaction regimes. We thus analyze the collapse of the hydrodynamic expansion after a sudden quench into the disorder potential of variable duration $\tau_{\text{on}}$. In addition to studying the hydrodynamic behavior, we have additionally examined the density response for all measurements along the crossover shown in the following and report it in Supplementary Note 2. In Fig. 2a, we compare the decay of the maximum achievable aspect ratio of hydrodynamic expansion of a mBEC ($1/k_{\text{F}}a = 1.04$) and a UFG at resonance ($1/k_{\text{F}}a = 0$) below the critical temperature as a function of $\tau_{\text{on}}$. All data are normalized to the maximum aspect ratio change experimentally obtained without the disorder, see Fig. 1c. Here, for the mBEC regime, the speckle laser power is set to 5 W, corresponding to $\bar{V}_{\text{dis}}^{\text{mBEC}}/\mu_{\text{mBEC}} \approx 1.6$ and 10 W for the UFG $\bar{V}_{\text{dis}}^{\text{UFG}}/\mu_{\text{UFG}} \approx 1.3$, ensuring approximately the same, strong disorder potential height in the two regimes. As seen in Fig. 2a, the maximum aspect ratio decreases after suddenly applying the disorder potential for a certain time $\tau_{\text{on}}$ approximately exponentially, indicating a collapse of quantum hydrodynamics and hence, long-range coherence. The

minimum aspect ratio reached in a steady state corresponds to the respective aspect ratio of a thermal gas in the mBEC regime or at resonance. We find that the UFG stabilizes at a higher value of the aspect ratio compared to the mBEC, which we attribute to higher interaction strength and hence, stronger classical hydrodynamics in this regime. As shown in the Methods, the change in aspect ratio depends on the trap frequencies rather than potential depth. Therefore, the difference in classical hydrodynamics cannot be due to the different masses of molecules compared to atoms. The half-life period $\tau_{1/2}$, i.e., the time at which the aspect ratio has collapsed to half of the initial value, of the UFG is shorter with $(7 \pm 1)$ μs than for the mBEC with $(11 \pm 2)$ μs . From studies in the mBEC regime[32], a relatively constant $\tau_{1/2}$ value was found for a broad range of interaction parameters outside the crossover $1/k_{\text{F}}a > 1$. The underlying mechanism explaining the time scales and numerical simulations of the phase evolution suggested the imprinting of a random local phase onto the mBEC by the disorder. The resulting local phase evolution destroys long-range phase coherence, and for a broad range of interaction strengths, it is only dependent on the properties of the disorder potential but not on interaction properties. In Fig. 2b, we show the $\tau_{1/2}$ values when entering the crossover. The clearly faster decay of long-range phase coherence indicates that the decay now does depend on interaction, in contrast to deep in the weakly interacting BEC regime ($1/k_{\text{F}}a \gg 1$). Hence, an additional mechanism suppressing long-range phase coherence is present in the crossover. To illustrate this, we compare the half-life period with a heuristic dephasing time $t_{\text{ph}} = \hbar/\Delta E$, which describes the time it takes for the two-particle wave function of the pair to accumulate a phase shift of unity due to a potential difference $\Delta E$ in the disorder field. A Feshbach molecule experiences a reduced potential difference compared to a more delocalized Cooper pair due to its smaller spatial extension. The decay of the half-life period by entering the crossover shows a qualitatively similar behavior as the decrease of the dephasing time, illustrating that a microscopic mechanism perturbing the pair structure could become relevant in the crossover.

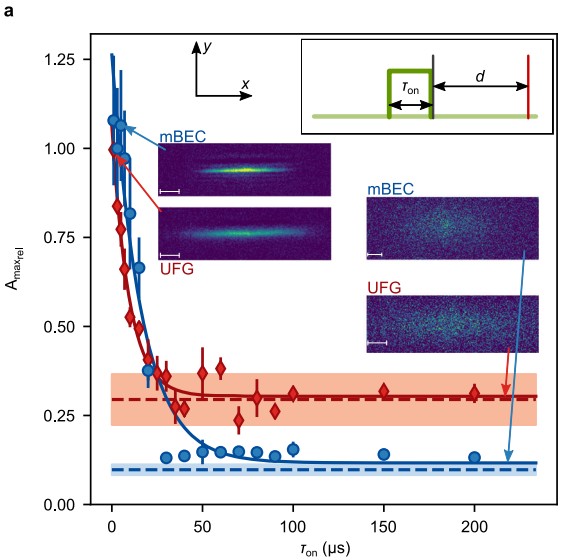

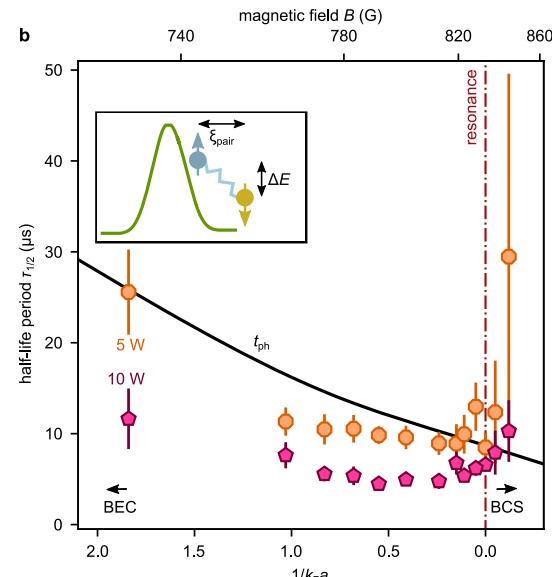

**Fig. 2 | Decay of quantum hydrodynamics. a** Decay of the relative maximum aspect ratio $A_{\max_{\mathrm{rel}}}$ (normalized to the maximum aspect ratio without disorder) as a function of the disorder-pulse duration $\tau_{\mathrm{on}}$ for a mBEC at 763.6 G ($1/k_F a = 1.04$, blue circles) and a UFG ($1/k_F a = 0$, red diamonds). The blue (red) dashed lines show independently measured aspect ratios for classical hydrodynamics in thermal gases at 763.6 G (832.2 G), i.e., including collisional hydrodynamics and its uncertainty as shaded areas (for more details see Supplementary Note 1C). The red and blue lines are exponential-decay fits to the data. Insets show the experimental sequence of disorder quench pulse (green), and subsequent expansion time $d$ and imaging (red vertical line) as well as absorption pictures of the two regimes for a

short and long disorder pulse. **b** Half-life period $\tau_{1/2}$ as a function of interaction parameter in the crossover for 5 W (orange octagons) and 10 W (pink pentagons) disorder laser power. In the crossover, the ability for hydrodynamic expansion decays significantly faster compared to interaction parameters $1/k_F a > 1$ far from the crossover[32]. The black line shows the calculated dephasing time $t_{\mathrm{ph}} = \hbar/\Delta E = \hbar\sigma_{x,y}/(\bar{V}_{\mathrm{dis}}\xi_{\mathrm{pair}})$ of the two atoms with opposite spin for $2\bar{V}_{\mathrm{dis}}$ for molecules (disorder laser power of 10 W). At this time scale, two paired atoms acquire a phase difference of unity in the disorder potential gradient, see inset sketch. Here, $\xi_{\mathrm{pair}}$ is taken from the calculation in Fig. 4b.

## Revival of quantum hydrodynamics

Complementary, we study the revival of the long-range phase coherence when the system is quenched out of a disordered potential. Experimentally, we adiabatically ramp up the speckle potential in a 50 ms linear ramp to avoid excitations in the cloud. The potential is held for 100 ms before suddenly switching off the disorder, see inset in Fig. 3a. After a varying hold time $\tau_{\mathrm{off}}$ without the disorder, the cloud is expanded for a time $d$ into the magnetic saddle potential to allow for hydrodynamic expansion before imaging. The two regimes of mBEC and UFG show strikingly different behavior, illustrated in Fig. 3a, b. While the long-range phase coherence for the mBEC recovers completely with an exponential time constant $\tau_{1/2}^{\mathrm{off}} = (20 \pm 3)$ms, the UFG does not fully revive. We show the maximum achievable value of the aspect ratio in Fig. 3c along the crossover without disorder and for two different disorder laser powers. Without disorder, we find a maximum aspect ratio that slightly increases for decreasing interaction parameter $1/k_F a$ in the crossover, until it rapidly decays for negative scattering length, i.e., on the BCS side of the crossover $1/k_F a < 0$. For the strongest disorder applied, the gas quenched out of the disorder behaves similarly to a classical gas for all interaction strengths. An interesting behavior is observed for intermediate disorder laser powers. Here, the gas can fully revive in the mBEC regime, which is consistent with previous studies[32]. For decreasing interaction parameter $1 > 1/(k_F a) > 0.5$, the maximum aspect ratio decays and approaches the classical limit until it increases again for $0.5 > 1/(k_F a) > 0$. However, for all interaction strengths in the crossover, quantum hydrodynamics does not revive, in stark contrast to the mBEC regime outside the crossover. This is even more remarkable since, for the equilibrium, weak-disorder phase diagram, the critical temperature in the mBEC regime is expected to be more strongly reduced with the disorder compared to the UFG[18,43].

In order to characterize the many-body state after the quench, we additionally measure the temperature increase for different disorder

ramp procedures, (see Methods). In Fig. 3d–f, we show the relative temperature increase for the limiting cases of an mBEC and a UFG for a fully adiabatic ramp, a disorder pulse (as used in Fig. 2), and rapid quench out of disorder (as used in Fig. 3a, c).

For the fully adiabatic ramp (Fig. 3d), neither mBEC nor UFG shows a significant increase in temperature. For a quench pulse, i.e., a quench into and a quench out-of the disorder (Fig. 3e), both gases are significantly heated and show a similar increase in temperature. While the relatively large error bars do not allow the identification of the speckle power when the gas is heated above the critical temperature, for the highest disorder potential, the UFG shows $T > T_c$. Interestingly, for an adiabatic loading into the disorder potential and a subsequent quench out of disorder, Fig. 3f, the UFG is heated more strongly than the mBEC, whereas for a speckle laser power of 10 W, the UFG is brought above the critical temperature, while the mBEC is mainly unaffected. The maximum thermal energy increase in Fig. 3f for the UFG is $\Delta E/E_F \approx 0.09$, which may be compared to the energy needed for pair breaking $2\Delta_{\mathrm{gap}} = 1.8E_F$[44]. The energy absorbed by the UFG from the quench thus brings the gas close to or above the critical temperature, and quantum hydrodynamic expansion breaks down.

In order to check if the temperature increase is the main reason for the collapse of quantum hydrodynamics, we study disorder quenches in an open system, where high-energetic particles can escape from the trap, effectively reducing the mean energy. This is achieved by reducing the depth of the optical dipole trap, allowing a certain fraction of atoms with the highest energy to escape. The relation between the fraction of particles lost and the potential depth is shown in Supplementary Note 3. We find that, even for losses of more than 60 %, the mBEC recondenses and shows a close-to-full revival of hydrodynamic expansion, and hence long-range coherence. By contrast, the UFG does not show a significant increase in quantum hydrodynamics for any particle loss.

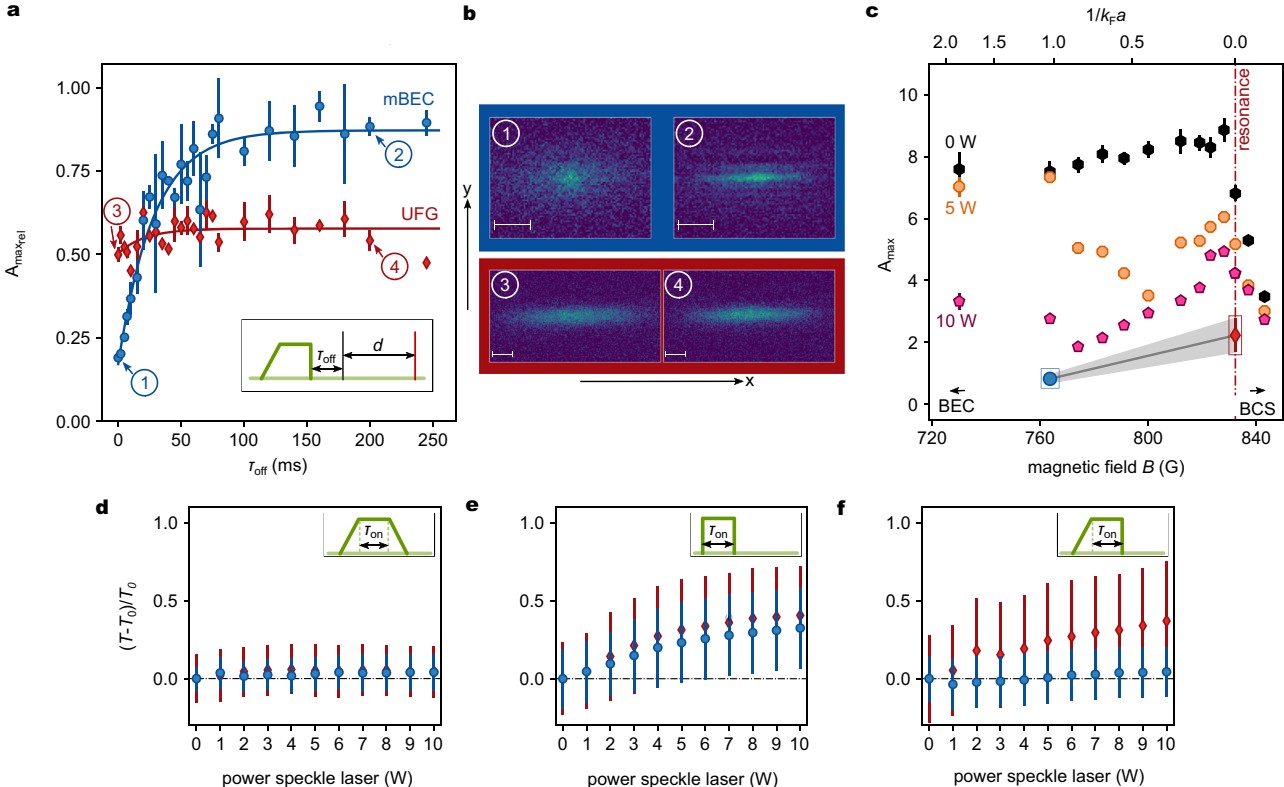

**Fig. 3 | Revival of quantum hydrodynamics. a** Relative revival of the maximum aspect ratio (normalized to the maximum aspect ratio without the disorder) $A_{\max_{\text{rel}}}$ as a function of the revival time $\tau_{\text{off}}$ in the mBEC regime (blue circles) and in the UFG (red diamonds). The red and blue lines are exponential fits. The inset shows the timing sequence. For the unitary (mBEC) regime, a speckle laser power of 10 W (5 W) is used to approach identical disorder potentials. **b** Absorption pictures for the measurement points indicated in panel (**a**). **c** Maximum absolute aspect ratio $A_{\max}$ after quenching out of disorder as a function of the interaction parameter $1/k_F a$. Dots are measurements for 5 W (orange octagons), 10 W (purple pentagons), and 0 W (black hexagons) disorder laser power. Each data point is an average of the three largest aspect ratios during the expansion, its uncertainty is the standard deviation of these three points. The blue (mBEC) and red (UFG) data points, marked with a rectangular box, indicate the independently measured aspect ratios of a thermal gas. The gray line is a guide to the eye between the two points, including its uncertainty. **d**–**f** Measured relative temperature increase for different disorder quench protocols shown in the insets, as a function of disorder power when applied to a mBEC (blue circles) or a UFG (red diamonds). The pulse duration is set to $\tau_{\text{on}} = 100$ ms. Each data point is an average of five repetitions, and their standard deviation is indicated as an error bar.

## Discussion

Our data suggests that even for a UFG initially adiabatically prepared to populate the ground state of a disorder potential, a quench out of disorder permanently destroys long-range quantum coherence. This is in stark contrast to the equilibrium expectation, where a UFG has so far been found to form a superfluid showing the largest critical velocity. The temperature measurements suggest the appearance of an additional absorption channel for the UFG, which is not present for the mBEC. This additional heating brings the UFG close to or above the critical point, and quantum hydrodynamic ceases.

A first intuitive understanding of the microscopic origin of this heating channel can be obtained by considering the microscopic pairing structure along the BEC-BCS crossover[11,45–47]. Figure 4a compares relevant energies of the quenched disorder potential with energy scales of the many-body system, especially the many-body gap $2\Delta_{\text{gap}}$ and the molecular binding energy $E_B = \hbar^2/(ma^2)$[34], in the BEC-BCS crossover. Calculations are done using experimental parameters, specifically total atom number $N_{\uparrow\downarrow} = 5.13 \times 10^5$ (average of experimental values) and trap frequencies $\omega_x, \omega_y, \omega_z = 2\pi \times (345, 23, 220)$ Hz. The Fermi energy is calculated according to eq. (2). Besides, the mean disorder potential strength $\bar{V}_{\text{dis}}$ is shown, and the correlation energy $E_\sigma = \hbar^2/(m\sigma^2)$[48] is calculated. These two quantities are different for atoms or molecules at the same laser power due to the difference in mass. The energy scale of the quench time of the speckle field is given by $E_{\text{quench}} = \hbar/t_{\text{on}}$, where the time to switch on the speckle field instantaneously is measured with $t_{\text{on}} = 2.26$ μs. Figure 4b shows the change of coherence lengths $\xi_{\text{pair}}$ and $\xi_{\text{phase}}$ of the interacting many-body system along the BEC-BCS crossover, see refs. 11,45–47. Deep in the mBEC regime, both differ significantly. The molecules are relatively small, and the healing length $\xi = \xi_{\text{phase}}$ is much larger, increasing for larger interaction parameters $1/(k_F a)$. Hence, $\xi_{\text{phase}} > \xi_{\text{pair}}$. Approaching the resonance, the healing length decreases, while the molecules become larger as the molecular bound state energy approaches the dissociation threshold. Upon entering the crossover $1/(k_F a) \approx 1$, the two length scales approximately coincide $\xi_{\text{phase}} = \xi_{\text{pair}}$. On resonance, the pair coherence length is larger than the phase coherence length. In the BCS regime, the two quantities scale the same and differ only by some constant factor. The coherence lengths allow for estimating the effect of local potential changes on the fermionic pair.

Comparing these scales with energy and length scales of the quenched disorder potential, we find that, first, deep in the mBEC regime, the pair size is much smaller than any length scale of the disorder potential. At the same time, the molecular binding energy is so large that no energy scale of the disorder is comparable. Thus, the disorder potential primarily affects the global wave function, which is perturbed at the length scale of the healing length and the energy scale of the chemical potential, leading to wave-like excitations of the many-body system. In the crossover, however, the pair

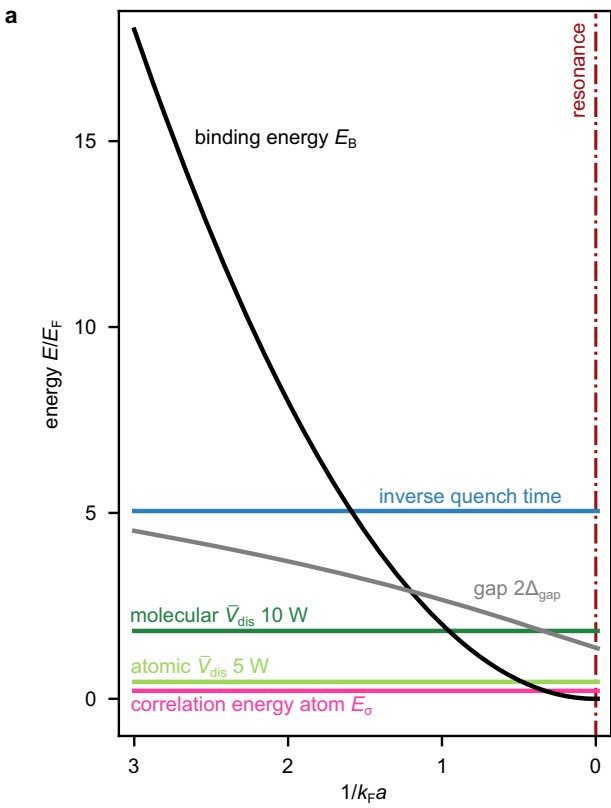

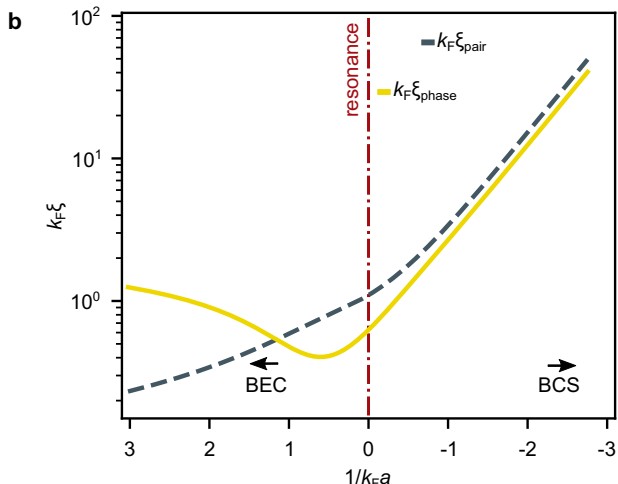

**Fig. 4 | Energy and length scales. a** Comparison of relevant energy scales along the crossover in units of the Fermi energy $E_F$ as a function of the interaction parameter $1/k_F a$. Binding energy and gap are calculated for the three-dimensional case at zero temperature according to refs. 11,45–47. Calculations of the further energies are done with experimental parameters. **b** Pair correlation length $k_F \xi_{pair}$ (gray dashed line) and phase coherence length $\xi_{phase}$ (yellow line) as a function of the interaction parameter $1/k_F a$, calculated for the three-dimensional case at zero temperature according to refs. 11,45–47.

inverse ramping time $E_{quench} = \hbar/t_{on}$, but also the correlation energy $E_\sigma = \hbar^2/(m\sigma^2)$[48] or mean potential strength $\bar{V}_{dis}$, become compatible to or larger than the many-body gap or the binding energy $E_B = \hbar^2/(ma^2)$[34] for sufficiently strong interaction. Hence, pair breaking can occur.

This microscopic matching of pair properties to the length and energy scales of the quenched disorder can explain the additional absorption channel for energy in the UFG. It brings the many-body system close to or even above the critical temperature, where quantum dynamics cease. For two-dimensional strongly disordered superconductors, it was shown in ref. 49 that the strongly disordered many-body system can form superconducting islands separated by an insulating sea, where the islands are not phase coherent. A fragmented Fermi gas, comprising unconnected islands of bound pairs in strong disorder, was proposed in ref. 24 to explain the experimentally observed density modulations. Our system is three-dimensional, and our disorder is formed by a repulsive potential far from the percolation threshold so the system is expected to be always fully connected. However, our observation might point toward a local dephasing of the UFG by strong disorder quenches, potentially at the level of individual pairs. It will be interesting in the future to see if the UFG quenched out of strong disorder might contain islands of Fermi pairs that connect to a smooth density but preserve phase differences that prevent global quantum hydrodynamic expansion.

Our studies of the stability of interacting Fermi superfluids along the BEC-BCS crossover in the presence of a strong time-dependent perturbation clearly show that the static properties of the disordered BEC-BCS crossover are very different from the observations obtained in this work for the time-dependent nonequilibrium case. In the future, it will be interesting to quantitatively map out the phase diagram of the disordered BEC-BCS crossover away from equilibrium. In addition, reducing the dimensionality of the gas and producing homogeneous gases in different dimensions will provide a deeper and more quantitative insight into the connection between the microscopic pairing mechanism and macroscopic nonequilibrium dynamics.

## Methods
### Quantum-gas preparation
We evaporate our laser-cooled samples at a magnetic field of 763.6 G ($a = 4509 a_0$, with the Bohr radius $a_0$) in the mBEC regime for all observed interacting regimes. During evaporation, the laser power of the optical-dipole trap (ODT) is lowered from 140 mW to 8 mW by two subsequent exponential power ramps during 4.38 s. This final laser power of 8 mW is relatively low and enables controlling atom losses through disorder quenches. After the evaporation step, the laser power is held for 250 ms to equilibrate the temperature of the cloud. Subsequently, the trap is compressed by increasing the optical-dipole-trap (ODT) laser power up to 80 mW during 300 ms. By varying the level of compression, the amount of losses due to the disorder field can be adjusted. No losses occur at a power of 80 mW. After the condensate is formed, the magnetic field is adiabatically ramped to the desired field for interaction control during 200 ms. The cloud is trapped in a combination of the ODT, created with a focused laser beam of a wavelength of 1070 nm and a magnetic saddle potential. Further, the magnetic field strength determines the scattering length and, therefore, the interacting regime between the two spin states. Before disturbing the cloud by quenching the disorder field, another holding time of 30 ms ensures that no excitations of the cloud are present after changing the magnetic field strength. The trap frequencies in the radial directions, $x$ and $z$, increase with the square root of the laser power of the ODT. The axial trap frequency, $y$ direction, is within the range used, independent of the laser power. Changing the magnetic field has a negligible influence on the trap frequency compared to the genuine frequency. The trap frequencies are

size increases and becomes of the order of the healing length, being smaller but of the order of the correlation length of the disorder, where the UFG shows the largest pair size of the order of $1/k_F$. As shown in the inset of Fig. 2b, local gradients of the disorder hence become relevant on the length scale of the pair. At the same time, the many-body gap, as well as the binding energy of the molecular state, decreases when approaching the resonance $1/k_F a \to 0$. Here, the energy scales of the time-dependent disorder, specifically the

$\omega_x, \omega_y, \omega_z = 2\pi \times (345, 23, 220)$ Hz for an ODT laser power of 80 mW and a magnetic field strength of 763.6 G.

### Disorder potential

A far-off resonant, blue-detuned laser with a wavelength of 532 nm generates the potential. The envelope of the laser is a Gaussian with waists of $(466 \pm 25)$ μm and $(414 \pm 25)$ μm along two orthogonal directions[50]. Moreover, the optical speckle potential is formed by shining the collimated laser beam through two successive speckle plates. An objective focuses the random phase pattern on the position of the atoms. Hence, anisotropic speckle grains with sizes $\sigma_{x,y} = 750$ nm, and $\sigma_z = 10$ μm are formed. The speckle plates are mounted in a motorized turntable; one of them is rotated by a certain angle after each measurement so that the interference potential changes in each experimental realization. We characterize the strength of the disorder potential $V_{dis}$ by the mean disorder potential $\bar{V}_{dis}$, which is the overall spatial average. For comparison, we may express the mean disorder potential in units of the unperturbed chemical potential. For the mBEC, the chemical potential[51]

$$\mu_{mBEC} = \frac{\hbar\bar{\omega}}{2}\left(15\frac{N_{\uparrow\downarrow}}{2}\frac{a_{dd}}{a_{ho}}\right)^{2/5} \qquad (3)$$

is established through the Gross-Pitaevskii equation, where $a_{dd} = 0.6a$[52] is the s-wave scattering length for molecules, $a_{ho} = \sqrt{\hbar/(2m\bar{\omega})}$ the harmonic oscillator length with the mass $m$ of the lithium atom. The chemical potential of the UFG[53]

$$\mu_{UFG} = \sqrt{1+\beta}E_F \qquad (4)$$

is proportional to the Fermi energy with the universal constant $\sqrt{1+\beta}$.

### Dependence of the expansion dynamics on interactions

An mBEC and a UFG show quantum hydrodynamic expansion[34]. In the case of an mBEC, a UFG, or a thermal gas, the in-situ aspect ratio is independent of the mass of the particles. The radii for all three cases individually depend on the mass, but it cancels when computing the aspect ratio. Instead, the aspect ratio depends on the trap frequencies. During expansion, the aspect ratio $\frac{R_x(t)}{R_y(t)} = \frac{b_x(t)}{b_y(t)}\frac{\omega_y}{\omega_x}$ changes according to scaling factors $b_i$

$$\ddot{b}_i - \frac{\omega_i^2}{b_i(b_xb_yb_z)^\gamma} = 0, \qquad (5)$$

which can be derived from the Euler equation[34]. The scaling factors are independent of the mass. The expansion of the mBEC and the UFG differ by the exponent gamma (BEC $\gamma = 1$, UFG $\gamma = 2/3$) since the difference of the chemical potential depends on the density in the two regimes ($\mu_{BEC} \propto n$, $\mu_{UFG} \propto n^{2/3}$)[54]. Moreover, the trap frequencies are identical for a Feshbach molecule or a single atom, and hence the aspect ratio during expansion is independent of the mass of the particles. A thermal gas shows a ballistic expansion with scaling factors also independent of the particle mass.

### Determination of the half-life period by expansion dynamics

For the decay and the revival of the long-range phase coherence (where the cloud fully recovers), the half-life period $\tau_{1/2}$ is determined similar to ref. 32. When plotting the aspect ratio as a function of the speckle-pulse length $\tau_{on}$ or the revival time $\tau_{off}$, it shows an exponential evolution to a steady state. This behavior is fitted with an exponential decay function (see Figs. 2a, 3a)

$$h(\tau) = a\,e^{-\gamma\tau} + o, \qquad (6)$$

with $\gamma$ is the decay constant and $o$ and $a$ are further fit values. Here, the half-life period is calculated via

$$\tau_{1/2} = \ln(2)/\gamma. \qquad (7)$$

The uncertainty of the half-life period is taken as the fit uncertainty.

### Temperature measurements

We have measured the relative temperature increase in the quantum gases for a mBEC at 763.6 G and a UFG at 832.2 G through the disorder potential for different disorder ramp procedures (see Fig. 3d–f) applying up to 10 W laser power creating the disorder field. As a reference, we measure the gas temperature $T_0$ without the disorder field. Three disorder ramping procedures are investigated. First (Fig. 3d), the disorder field is adiabatically introduced by a linear ramp during a 50 ms, held for 100 ms, and subsequently switched off at 50 ms by a linear ramp. Second (Fig. 3e), the disorder is rapidly switched on (switching time of $t_{on} = 2.26$ μs) for a rectangular pulse for a duration of duration $\tau_{on} = 100$ ms. Lastly (Fig. 3f), the disorder field is adiabatically introduced by a linear ramp during a 50 ms, and subsequently, the power is held for 100 ms, before the field is suddenly switched off. After a holding time of 100 ms, the magnetic field is adiabatically swept to 680 G. Subsequently, the cloud is imaged in situ via absorption imaging. At this magnetic field strength, a bimodal fit to the density profiles allows the extraction of the temperature of the cloud.

## Data availability

All data of the figures in the manuscript are available in a Zenodo repository, Ref. 55, https://doi.org/10.5281/zenodo.13292670.

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

## Acknowledgements

We thank Carlos A. R. Sá de Melo, Henning Moritz, Giuliano Orso, Giacomo Roati, Wilhelm Zwerger, Benjamin Nagler, and Alexander Guthmann for helpful discussions and Eloisa Cuestas for carefully reading the manuscript. This work was supported by the German Research Foundation (DFG) via the Collaborative Research Center Sonderforschungsbereich SFB/TR185 (Project 277625399). J.K. was supported by the Max Planck Graduate Center with the Johannes Gutenberg-Universität Mainz. We acknowledge the help by Valeriya Mikhaylova with the graphical implementation of Fig. 1a.

## Author contributions

A.W. conceived and supervised the research. J.K. took and analyzed the experimental data and calculated the theoretical estimations. S.B. and F.L. helped run the experimental apparatus and collect data. All authors contributed to the interpretation of the data, writing of the manuscript, and critical feedback.

## Funding

## Competing interests

The authors declare no competing interests.
