## [Peer Review File · Nature Communications]

Stability and sensitivity of interacting fermionic superfluids to quenched disorderREVIEWER COMMENTS

Reviewer #1 (Remarks to the Author):

In the manuscript "Stability and sensitivity of interacting fermionic superfluids to quenched disorder" by Jennifer Koch et al.,

the Author experimentally investigate the expansion of an interacting Fermi gas of Li atoms in the repulsive side of the BEC-BCS crossover in the presence of disorder potential realized with a speckle pattern.

In particular, they concentrate their investigation on the dynamical response of the system to a sudden switch on or off of the disorder. The long-range coherence of the fermionic superfluid is monitored by measuring the hydrodynamic expansion through the aspect ratio of the cloud.

The main and highly relevant, observation is that, while in the case of static disorder (as predicted by the Anderson theorem), the superfluid formed by localized pairs, i.e. the molecular BEC, is strongly affected than that formed by delocalized pairs, i.e. the unitary or BCS Fermi gas, here the response to a time-dependent disorder is stronger on the unitary side of the crossover.

Moreover, the molecular BEC is affected by lower heating, and once out of the disorder, it restores its hydrodynamics.

This observations indicates that the response of a superfluid formed by pairs is strongly affected by the pairs nature.

As clearly stated in the last Discussion section, "the temperature measurements suggest the appearance of an additional absorption channel for the UFG, which is not present for the mBEC."

The Authors provide an intuitive explanation based on the microscopic structure of the pairs in the two different regimes, along with a comparison between different energy and length scales (phase and pair coherence length). This justification appears elegant and simple, effectively explaining the observed behavior.

In the conclusion, interesting prospects and directions for future investigation are outlined. Additionally, the potential effect of the spatial correlation length of the speckle potential may warrant further study.

The work is certainly original, interesting, clear and well written, and I believe it meets the criteria for publication in Nature Communications. This study represents an important advancement in understanding the general problem of the physics of disordered systems and the comprehending the diverse microscopic nature of interacting superfluids.

For all these reasons, I think that this manuscript deserve publication one the Authors have considered the following list of observations/suggestions:

- As the expansion is "initiated by... releasing the gas in a magnetic trap" it would be helpful to know the parameters characterizing this trap without the necessity to refer to [31].
- The temporal sequence of the disorder quench is clearly explained by the inset of fig.2a. I suggest referring to this grappò in the text from the outset.
- In fig.1b it would be useful to see the scale for the y-axis (aspect ratio). Additionally, regarding fig.1, it seems to me that the aspect ratio A is not defined. It would suffice to change line 121 in the text from "... the change of aspect ratio can be ..." to "... the change of aspect ratio A can be ...".
- In fig1c, the thicker continuous line is not described. Is it the expected behavior of A from eq.D1 in Appendix B?. The same comment applies to the dashed line in fig1d.
- I suggest to better stress that this work is an extension of the study reported in ref.17, where only the mBEC case was investigated.
- In line 371, there is a typo: "entering" instead of "entering".

- I suggest The Author to add references on previous experimental works on strongly interacting fermions in random potential as Phys. Rev. Lett. 110, 100601 (2014) and Phys. Rev. Lett. 115, 045302 (2015).

Reviewer #2 (Remarks to the Author):

The paper deals with the effect of a quench of the disorder in the mBEC to UFG crossover. It takes as a main observable the ability of the gas to exhibit anisotropic (hydrodynamic) expansion which is a signature of an initial phase coherence of the gas. Two experimental procedures are studied, a sudden increase of disorder for a given time and a slow addition of disorder prior to a quench to no disorder. The disorder is a speckle pattern with strong amplitude, comparable to the chemical potential. The introduction and the selling point of the article focuses on the fact that the reaction of a UFG to a disorder quench is much stronger than the one of a BEC, whereas the situation is inverted for static disorder. Globally, the results are well presented. They are mostly experimental observations and are not directly supported by theory. I consider that it is not a weak point to be ahead of any theory in this complex regime. However, I find that there are shortcomings in the interpretation and I would like the authors to consider some comments before I can really advise publication.

-The selling point of the paper is the difference between quench and static disorder. In particular, the authors mention in the introduction the Anderson theorem or the fact that the superfluid velocity is larger in the case of strong interaction. These arguments are valid in the weak disorder limit. This is not the regime experimentally studied in the paper and this can be at the origin of the 'surprising' difference to naive expectations. This important difference of strong disorder should be made clear.

-In the introduction, the regime of static disorder in the mBEC-BCS crossover seems to have been only studied theoretically. Is this true ?

-In the last sentence of the abstract, I do not understand 'where the local dephasing of pairs appears to be a crucial aspect'

-Please give the scattering lengths at 763 G, 832 G and 680 G. The value of k_F and density would be nice as well.

-In the on-off quench experiment, it is stated that the half-life period of the coherence was measured (in a previous paper) to be independent from the interaction in the mBEC regime. In the paper, such as independence is not visible. For what values of $k_F a$ was this measured ? It would be nice to really see when the half-life period of the coherence start to decrease. This would indicate the place where additional effects have to be taken into account. Second, close to the resonance, there is a slight increase of the half-life period. This result is not commented in the paper. Is it meaningful or is it linked to the fact that anisotropic expansions are never observed for $a < 0$? A comment would be welcome.

-At the end of the caption of figure 2, there is an estimate of the dephasing time of two atoms of a pair. What can one conclude from this time ? It does not really follow the data. Does it mean that the pair will be broken on such a time scale ? Why comparing to such a time if the dominant source of decoherence is a global phase shift for pairs (as it is the process put forward on the BEC side) ? In brief, some more explanations are needed. Can one also get a similar time scale by estimating the differential acceleration for two atoms from the same pairs ?

-The paper emphasizes the coherence measurements and in particular the fact that the UFG does not exhibit full revival of the coherence although the mBEC does. The authors are not making a direct link

to the fact that there is more heating, in particular because they claim that the amount of energy is not sufficient to break the pairs. I believe that this is a misinterpretation. The temperature is measured at 680 G in the weakly interacting mBEC regime (after an adiabatic magnetic field ramp). One should then compare the entropy per particle to find out the temperature of the gas in the crossover and compared this temperature the value of T_c . If one consider an increase of temperature to $0.4T_F$, and according to the reference 25, the gas is simply not far from the transition temperature at resonance. This would explain the incomplete revival of the coherence. On the contrary, the mBEC shows very little heating and thus exhibit a revival of coherence after thermalization. It seems to me that there is no mystery once the temperature increase has been measured

-The most interesting result of the paper may be the measurement of the temperature increase (or more the entropy increase after thermalization before the change of magnetic field) in the different cases. For the mBEC case, I believe that the behavior can be probably understood. When the disorder is suddenly turned on, the energy per particle increases by $\sim V$, the disorder strength. When the disorder is removed, the particles do not occupy the high disorder regions, and the energy increase is very reduced. This may be captured in the local density approximation. In the UFG case, it is more difficult, because the healing length and the size of the pairs are of the order of the correlation length of the disorder. In particular, in the presence of disorder, one can anticipate an effect on the pairs when suddenly removing the disorder. This may give information of the structure of the gas in the presence of disorder. Theory would be needed to quantitatively understand the effect. As mentioned in the paper, the temperature results suggest a direct effect of disorder on the pairs. I would say that pair breaking is likely in the quench, oppositely to what the authors are claiming.

-According to my previous remarks, I would put more emphasis on the temperature measurements (maybe extended to the full crossover) than on the coherence measurements. In fact, there is no real conclusion from the decoherence half-time measurement, nor from the revival times.

-The part on particle loss is not very convincing to me and does not seem to bring so much to the results. In particular, I do not see how one can make a statement for the revival experiment since the temperature increase is measured to be very different. This part seems to me unnecessarily. It could be put in the appendix only. In my mind, the fact that molecules feel twice the speckle potential as compared to atoms does not need to be verified.

We thank all the Reviewers for their detailed and constructive criticisms. We are glad to see that both reviewers see the novelty and future perspective in our work. We have considered all the points raised in the reports and modified the manuscript accordingly.

Reply to Referee 1:

1) *As the expansion is “initiated by... releasing the gas in a magnetic trap” it would be helpful to know the parameters characterizing this trap without the necessity to refer to [31].*

We have added more details of the magnetic trap in the main text, as it is a saddle potential configuration, which is confining in the x - y plane and anti-confining in z direction. As characterizing parameters, we add the trap frequencies of the optical-magnetic trap also in the main text of the manuscript at the beginning of the *Experimental realization*. The trap frequencies were prior only mentioned in the Methods.

2) *The temporal sequence of the disorder quench is clearly explained by the inset of fig.2a. I suggest referring to this graph in the text from the outset.*

We have implemented the link to the inset at the end of the experimental realization section, where the disorder realizations are described. In addition, we refer to the inset in Fig. 3a), where the disorder potential is rapidly switched off.

3) *In fig.1b it would be useful to see the scale for the y-axis (aspect ratio). Additionally, regarding fig.1, it seems to me that the aspect ratio A is not defined. It would suffice to change line 121 in the text from “... the change of aspect ratio can be ...” to “... the change of aspect ratio A can be ...”.*

We have added a scaling with logarithmic scale on the y-axis for Fig. 1b). To do so, we have extended the length of the graph in the y-direction. In order to achieve an optimal alignment of the panels for Fig. 1 with the new size of Fig. 1b), we have changed their arrangement. For completeness and better clarity, we have added the definition A to the suggested sentence.

4) *In fig1c, the thicker continuous line is not described. Is it the expected behavior of A from eq.D1 in Appendix B?. The same comment applies to the dashed line in fig1d.*

We have added the missing description. The two lines correspond to the computed ideal time evolution also shown in Fig. 1b). This corresponds exactly to the description in Appendix D, eq. (D1) (now eq. (6) in the new manuscript) describes the change of the scaling factors over time, which enter into the aspect ratio as follows $A = \frac{R_x(t)}{R_y(t)} = \frac{b_x(t)}{b_y(t)}$.

5) *I suggest to better stress that this work is an extension of the study reported in ref.17,*

where only the mBEC case was investigated.

Previously, we had referred to the previous study [17] in lines 115-118 with *While initial studies [17] have been performed deep in the molecular BEC (mBEC) regime, we focus on the markedly different stability and sensitivity of a mBEC and a UFG, respectively.* To further clarify that this work is an extension of our previous study, we have changed that statement to: *In a previous work [33], performed with the same experimental apparatus, we focused on investigations deep in the molecular BEC (mBEC) regime. Here, we extend this study by focussing on the markedly different stability and sensitivity of an mBEC compared to a UFG.*

6) *In line 371, there is a typo: "enterning" instead of "entering".*

We have corrected the typo.

7) *I suggest The Author to add references on previous experimental works on strongly interacting fermions in random potential as Phys. Rev. Lett. 110, 100601 (2014) and Phys. Rev. Lett. 115, 045302 (2015).*

We have added both references to the introduction of the manuscript as an example of already performed experimental work of strongly interacting fermions in random potentials and the disorder-induced transition from a superfluid to a normal fluid, respectively.

Reply to Referee 2:

1) *The selling point of the paper is the difference between quench and static disorder. In particular, the authors mention in the introduction the Anderson theorem or the fact that the superfluid velocity is larger in the case of strong interaction. These arguments are valid in the weak disorder limit. This is not the regime experimentally studied in the paper and this can be at the origin of the 'surprising' difference to naive expectations. This important difference of strong disorder should be made clear.*

We agree with the reviewer that is an important aspect. While we had already emphasized in several sections of the manuscript that the disorder is a strong perturbation, we have added the following part to the manuscript at the end of the second paragraph: "Importantly, most studies have considered the weak disorder regime, and theoretical investigations of strong-disorder systems beyond perturbation theory are just emerging [New J. Phys. 22 033021 (2020)]."

2) *In the introduction, the regime of static disorder in the mBEC-BCS crossover seems to have been only studied theoretically. Is this true ?*

Systematic experimental studies are scarce, and important general statement are concluded from theoretical approaches. There are studies either in the mBEC regime or for a strongly-interacting Fermi gas. We have added references suggested by the first reviewer. Additionally, we have added a previous work of ours measuring dipole oscillations of fermionic quantum gases along the BEC-BCS crossover in a disordered potential [Phys. Rev. A 101, 053633 (2020)]. We are not aware of other experimental studies of the disordered BEC-BCS crossover.

3) *In the last sentence of the abstract, I do not understand 'where the local dephasing of pairs appears to be a crucial aspect'*

We have removed this sentence from the abstract and explain this in more detail in the discussion. It has been predicted that in strong disorder, the many-body system can comprise islands of superfluids with different phases (see [12]). The limit at smaller length scales is a gas of pairs, where each pair has a different phase, possibly imprinted by a short length-scale disorder potential. Since we find that the energy absorption is not sufficient to break all the pairs, an interesting question is how microscopically the superfluid is affected when it is brought close to or above the critical temperature.

4) *Please give the scattering lengths at 763 G, 832 G and 680 G. The value of k_F and density would be nice as well.*

We have created a table with the relevant values in the Method section *Quantum-gas preparation*.

5) In the on-off quench experiment, it is stated that the half-life period of the coherence was measured (in a previous paper) to be independent from the interaction in the mBEC regime. In the paper, such as independence is not visible. For what values of $k_F a$ was this measured? It would be nice to really see when the half-life period of the coherence start to decrease. This would indicate the place where additional effects have to be taken into account. Second, close to the resonance, there is a slight increase of the half-life period. This result is not commented in the paper. Is it meaningful or is it linked to the fact that anisotropic expansions are never observed for $a < 0$? A comment would be welcome.

In previous work [PNAS119, e2111078118 (2022)], the hydrodynamic expansion was studied for disorder quenches in the BEC regime. For quenches out of disorder (Fig. 4F in PNAS119, e2111078118 (2022)), we find no significant change of the half-life time on interaction for gas parameters $n_0 a^3$ between 0.4×10^{-2} and 6×10^{-2} , where n_0 is the center density, and a the scattering length. (The detailed values of the gas parameters, scattering lengths and alike at the different magnetic field values are given in Table S1.) The response for quenches into disorder is shown in the supplemental material, Fig. S3. The closest magnetic field to the Feshbach resonance is 763.6 G, which connects to the data point farthest away from the resonance in our submitted manuscript.

We conclude from the new data that a significant change in behavior occurs around $1/k_F a \approx 1$. First, this is the range where the half-life period drops from $30 \dots 40 \mu\text{s}$ to $\leq 10 \mu\text{s}$. Second, as can be seen from Fig. 3c, this is the range when the hydrodynamic expansion of the quantum gas cannot recover from the disorder quench.

Concerning the increase of half-life period close to resonance, we do not know the reason. As the referee correctly states, there is no quantum hydrodynamic expansion observed for negative scattering length ($a < 0$), as illustrated in the black data points in Fig. 3c. Therefore, the decay time on the BCS side cannot be directly compared to the BEC side. However, as Fig. 3c shows, there is still a difference in the amplitude of hydrodynamic expansion for the BCS side close to resonance, hence the disorder has an effect. But from our data, we cannot draw any conclusion about the microscopic mechanism.

6) At the end of the caption of figure 2, there is an estimate of the dephasing time of two atoms of a pair. What can one conclude from this time? It does not really follow the data. Does it mean that the pair will be broken on such a time scale? Why comparing to such a time if the dominant source of decoherence is a global phase shift for pairs (as it is the process put forward on the BEC side)? In brief, some more explanations are needed. Can one also get a similar time scale by estimating the differential acceleration for two atoms from the same pairs?

The experimental observation shows that in the crossover the gas response is different from the response in the BEC regime. In particular, the global phase shift as put forward in the BEC regime cannot explain this behavior, as then all time scales would depend on the speckle potential but not on the atomic interaction properties. This clearly calls for other or

additional mechanisms dominating in the crossover.

The estimate of the dephasing time takes the idea of a phase shift from the BEC side (where the healing length is the dominant length scale) and applies this to smaller length scales, as in the crossover, where the pair size becomes the dominant length scale. The fact that this yields an order-of-magnitude agreement and a qualitatively similar curve in the crossover indicates in our view that a mechanism acting microscopically at the level of the pair size could come into play.

The dephasing describes the relative phase shift of an atom pair due to the potential gradient in the disorder field. The black line shows the dephasing time, i.e. the time during which the atoms of the pair accumulate a relative phase of unity. This depends on the energy difference the two atoms perceive, depending on the disorder strength, the grain size and the pair size. The pair size varies along the crossover: a Feshbach molecule is spatially small in contrast to a Cooper pair. Therefore, the potential difference and hence relative phase experienced by a Feshbach molecule in the disorder field is on average smaller than for a Cooper pair due to the pair size. Hence, the dephasing time ($t_{\text{ph}} = \hbar/\Delta E$) is larger. The spatial phase gradient accumulated by the pair could maybe drive an acceleration of the atoms to opposite directions, potentially leading to pair breaking. The aim of this comparison, however, was to illustrate that microscopic energy scales become relevant.

The alternative time scale the reviewer suggests certainly yields a similar time scale, because the pair size is of the order of the correlation length, and the observed time to re-establish a smooth density distribution is of the order of the time when hydrodynamic expansion ceases. However, this mechanism applies either a classical picture (particle-like atoms accelerated by potential gradients emerging due to disorder switching), or for a quantum mechanical wave, first a phase gradient of the two-particle wave function has to be established in order to accelerate the two particles away from each other. What we estimate is precisely the time scale on which this gradient emerges.

We have added further clarifications to the paragraph before "Revival of quantum hydrodynamics", reading "Hence, we compare the half-life period with a heuristic dephasing time $t_{\text{ph}} = \hbar/\Delta E$, which describes time it takes for the two-particle wave function of the pair to accumulate a phase shift of unity due to a potential difference ΔE in the disorder field. A Feshbach molecule experiences a reduced potential difference compared to a more delocalized Cooper pair due to its smaller spatial extension. The decay of the half-life period by entering the crossover shows a qualitative similar behavior as the decrease of the dephasing time, illustrating that a microscopic mechanism perturbing the pair structure could become relevant in the crossover."

7) The paper emphasizes the coherence measurements and in particular the fact that the UFG does not exhibit full revival of the coherence although the mBEC does. The authors are not making a direct link to the fact that there is more heating, in particular because they claim that the amount of energy is not sufficient to break the pairs. I believe that this is a misinterpretation. The temperature is measured at 680 G in the weakly interacting mBEC regime (after an adiabatic magnetic field ramp). One should then compare the entropy per particle to find out the temperature of the gas in the crossover and compared this temperature

the value of T_c . If one consider an increase of temperature to $0.4T_F$, and according to the reference 25, the gas is simply not far from the transition temperature at resonance. This would explain the incomplete revival of the coherence. On the contrary, the mBEC shows very little heating and thus exhibit a revival of coherence after thermalization. It seems to me that there is no mystery once the temperature increase has been measured

We fully agree with the reviewer that heating the gas above the critical temperature is responsible for the break-down of quantum hydrodynamics. We cannot conclude from the data, however, that for all the cases where hydrodynamics is absent, the gas is above T_c . Also, we cannot conclude that for these cases all the pairs have been broken.

In fact, the temperature estimates are based on the comparison of the entropy per atom as in Ref. [Phys. Rev. Lett. 95, 260405 (2005)] and Ref. [Science 335, 563 (2012)]. For the highest disorder strength (10W laser power), the quench results in an increase of T/T_F by approximately 0.2, with an initial T/T_F below 0.17. Given that $T_c/T_F = 0.167$, it can be concluded that while the gas was superfluid prior to the quench, for such high disorder strengths, T_c is exceeded and no revival takes place. We have clarified this in the manuscript. A different question relates the origin of the additional heating. We think that this can only be explained by mechanisms affecting the pairs rather than the global wave function.

8) The most interesting result of the paper may be the measurement of the temperature increase (or more the entropy increase after thermalization before the change of magnetic field) in the different cases. For the mBEC case, I believe that the behavior can be probably understood. When the disorder is suddenly turned on, the energy per particle increases by V , the disorder strength. When the disorder is removed, the particles do not occupy the high disorder regions, and the energy increase is very reduced. This may be captured in the local density approximation. In the UFG case, it is more difficult, because the healing length and the size of the pairs are of the order of the correlation length of the disorder. In particular, in the presence of disorder, one can anticipate an effect on the pairs when suddenly removing the disorder. This may give information of the structure of the gas in the presence of disorder. Theory would be needed to quantitatively understand the effect. As mentioned in the paper, the temperature results suggest a direct effect of disorder on the pairs. I would say that pair breaking is likely in the quench, oppositely to what the authors are claiming.

We completely agree with the statement of the referee. A detailed theoretical analysis is necessary together with more experimental work in a well-defined setting in order to make reliable statements. Due to the complexity of the experimental system, we are in a very difficult theoretical regime. We work together with Giuliano Orso and Carlos Sa de Melo on this problem. With G. Orso, we have successfully understood the experimental data in the BEC regime from numerical results.

Some complications occur also from non-ideal experimental conditions, such as inhomogeneous density distributions, the previous inability to probe the pairs, and the quench protocol comprising a broad spectrum of excitation frequencies. We plan to overcome these issues providing data from a uniform density sample through modulation spectroscopy to obtain

a full spectrum, where the macroscopic hydrodynamic data is paralleled by measurements of the microscopic pair fraction. This, however, will require significant rebuilding of the experimental apparatus.

Concerning the point of pair breaking, we have indeed softened our statement. We agree that the thermal energy increase does not allow a statement on the number of pairs broken.

9) According to my previous remarks, I would put more emphasis on the temperature measurements (maybe extended to the full crossover) than on the coherence measurements. In fact, there is no real conclusion from the decoherence half-time measurement, nor from the revival times.

We agree that the strong difference between the heating of the UFG compared to the BEC is a central issue, however, the error bars of the relative temperature change are relatively large. For this reason, we have deliberately taken the temperature data only at limiting values of the interaction strength. As mentioned above, we plan on revisiting this issue with a better controlled experimental system, where we additionally plan on using different measurement methods for temperature to reduce errors and provide more precise statements. In addition, we want to investigate the reason for the temperature increase with the support of our colleagues from theory.

We have not reported on the revival time in our manuscript, because the UFG does not revive. Therefore, no revival time can be determined and a comparison with the BEC regime is not possible.

However, there is an important conclusion of the decoherence half-time measurement: its value changes in the crossover compared to the BEC regime, i.e. it becomes interaction dependent. The previous explanation for decay of quantum hydrodynamics is fully independent of the interaction parameter. Hence, this data shows that a new mechanism must be taken into account. We have better emphasized this in the paragraph before the section "Revival of quantum hydrodynamics".

10) The part on particle loss is not very convincing to me and does not seem to bring so much to the results. In particular, I do not see how one can make a statement for the revival experiment since the temperature increase is measured to be very different. This part seems to me unnecessarily. It could be put in the appendix only. In my mind, the fact that molecules feel twice the speckle potential as compared to atoms does not need to be verified.

We have followed the suggestion of the reviewer and moved this part to the Methods. The conclusion from this concerns rather the ability of a BEC to re-establish quantum properties. The question of molecules experiencing twice the potential of atoms is certainly obvious. It is not so obvious if a Fermi pair acts as one molecule or as two independent atoms. In literature, implicitly one or the other has been assumed. For example, if some tunneling dynamics is recorded across the BEC-BCS crossover and plotted for one constant laser power of a barrier, then implicitly it has been assumed that the Fermi pair acts molecule-like, experiencing the same potential as the molecule. Our quench data shows that this is not the case for our work.

REVIEWERS' COMMENTS

Reviewer #2 (Remarks to the Author):

I have read the resubmitted version of the manuscript with interest. The authors have taken the reports into account and have clarified both the introduction, the presentation and the interpretation of the results. I now recommend publication of the manuscript in nature communications. The work clearly bring new results and insights as compared to previous studies.

I have some minor comments that could be clarified:

-What is the precise definition of σ for the speckle potential used in the paper ? There are different definitions in the literature.

-In fig. 2, they is the curve of the calculated dephasing time assumes a pair size in the BEC crossover. I guess the values from figure 4 has been used. I could be clarified.

-In table 1 in the supplementary material, the peak density varies significantly whereas k_F is not. Is there an error ?

We thank the Reviewer for their detailed and constructive criticisms. We have considered all the points raised in the report and modified the manuscript accordingly.

Reply to Referee 2:

1) What is the precise definition of sigma for the speckle potential used in the paper? There are different definitions in the literature.

The correlation length we use is the $1/e$ -width of the speckle potential's autocorrelation function. We have added this definition and a citation to the manuscript.

2) In fig. 2, they is the curve of the calculated dephasing time assumes a pair size in the BEC crossover. I guess the values from figure 4 has been used. I could be clarified.

Indeed, the pair size used in Fig. 2 is the one computed and shown in Fig. 4. We have added an explanation to the caption of Fig. 2.

3) In table 1 in the supplementary material, the peak density varies significantly whereas k_F is not. Is there an error?

This is not an error. The wave vector is computed from the Fermi energy $k_F = \sqrt{2mE_F}/\hbar$. The Fermi energy depends on the total atom number and the trap frequency $E_F = (6N)^{1/3}\hbar\bar{\omega}$. The dependence of the wave vector on total atom number is very weak due to the high root drawn from the atom number. Moreover, the density given is the peak density and rather reflects the changing quantum state from a BEC (with high central density) to a unitary Fermi gas. We have added more explanation how the numbers are determined to the caption of Table 1.